# Socioeconomic risk factors for fatal opioid overdoses in the United States: Findings from the Mortality Disparities in American Communities Study (MDAC)

**Sean F. Altekruse** [1]*, **Candace M. Cosgrove** [2], **William C. Altekruse** [3], **Richard A. Jenkins** [4], **Carlos Blanco** [4]

**1** Division of Cardiovascular Sciences, National Heart, Lung, and Blood Institute, National Institutes of Health, Bethesda, Maryland, United States of America, **2** Center for Administrative Records Research and Applications, U.S. Bureau of the Census, Suitland, Maryland, United States of America, **3** Division of Translational Research, National Institute of Mental Health, National Institutes of Health, Bethesda, Maryland, United States of America, **4** Division of Epidemiology, Services, and Prevention Research, National Institute on Drug Abuse, National Institutes of Health, Bethesda, Maryland, United States of America

* altekrusesf@mail.nih.gov

**Data Availability Statement:** Minimal data for the present MDAC analysis were comprised of the following variables: cause of death, age, race,

## Abstract

### Background

Understanding relationships between individual-level demographic, socioeconomic status (SES) and U.S. opioid fatalities can inform interventions in response to this crisis.

### Methods

The Mortality Disparities in American Community Study (MDAC) links nearly 4 million 2008 American Community Survey responses to the 2008–2015 National Death Index. Univariate and multivariable models were used to estimate opioid overdose fatality hazard ratios (HR) and 95% confidence intervals (CI).

### Results

Opioid overdose was an overrepresented cause of death among people 10 to 59 years of age. In multivariable analysis, compared to Hispanics, Whites and American Indians/Alaska Natives had elevated risk (HR = 2.52, CI:2.21–2.88) and (HR = 1.88, CI:1.35–2.62), respectively. Compared to women, men were at-risk (HR = 1.61, CI:1.50–1.72). People who were disabled were at higher risk than those who were not (HR = 2.80, CI:2.59–3.03). Risk was higher among widowed than married (HR = 2.44, CI:2.03–2.95) and unemployed than employed individuals (HR = 2.46, CI:2.17–2.79). Compared to adults with graduate degrees, those with high school only were at-risk (HR = 2.48, CI:2.00–3.06). Citizens were more likely than noncitizens to die from this cause (HR = 4.62, CI:3.48–6.14). Compared to people who owned homes with mortgages, those who rented had higher HRs (HR = 1.36, CI:1.25–1.48). Non-rural residents had higher risk than rural residents (HR = 1.46, CI:1.34, 1.59). Compared to respective referent groups, people without health insurance (HR = 1.30,

Hispanic ethnicity, sex, disability, marital status, employment status, educational attainment, citizenship, housing tenure, rurality, health insurance status, incarceration, household poverty, and Census Division. MDAC data cannot be shared publically in accordance with U.S. Code Title 13 privacy protection requirements. Data are, however available on a need to know basis via submission of a request to the MDAC Steering Committee as outlined on the MDAC website: www.census.gov/mdac. Need to know justifications include analysis for publication, validation of prior publication findings, or extension of previous research. There are two principal methods available to access the full MDAC dataset. The authors of the present study accessed the MDAC dataset in coordination with a Census Bureau analyst. The other option is to independently access the data as a Census Bureau Special Sworn Status investigator, at a Federal Statistical Research Data Center (FSRDC). FSRDC information is found at the website: www.census.gov/fsrdc. The authors of the present study had no special access privileges in accessing MDAC which other interested researchers would not have.

**Funding:** The authors received no specific funding for this work. MDAC is funded by National Heart, Lung, and Blood Institute and National Institute on Aging agreements with the Census Bureau. The National Center for Health Statistics performed the National Death Index match and the Center for Medicare and Medicaid Services performed the CMS match.

**Competing interests:** The authors have declared that no competing interests exist.

CI:1.20–1.41) and people who were incarcerated were more likely to die from opioid overdoses (HR = 2.70, CI:1.91–3.81). Compared to people living in households at least five-times above the poverty line, people who lived in poverty were more likely to die from this cause (HR = 1.36, CI:1.20–1.54). Compared to people living in West North Central states, HRs were highest among those in South Atlantic (HR = 1.29, CI:1.11, 1.50) and Mountain states (HR = 1.58, CI:1.33, 1.88).

## Discussion

Opioid fatality was associated with indicators of low SES. The findings may help to target prevention, treatment and rehabilitation efforts to vulnerable groups.

## Introduction

In the United States fatal drug overdose rates more than tripled from 1999 to 2017. Opioid overdoses were by far the largest contributor to the 70,237 fatal drug overdoses in the United States during 2017. [1] The U.S. opioid epidemic has evolved over time. [2] In the early 1980s, opioids were primarily prescribed for acute pain, and a substantial fraction of drug-related deaths were attributed to the diversion to opioids to non-medical uses. A decade later, in response to perceived undertreatment, the practice of prescribing opioids for chronic pain management gained increasing favor. By 2000, this trend accelerated after the withdrawal of popular prescription nonopioid analgesics due to cardiovascular risks and because of concerns about acetaminophen toxicity. Circa 2010, combined opioid analgesic and heroin use was reported with increased frequency. By the late 2010s, potent products such as fentanyl and its analogs were increasingly reported in counterfeit pills and street drugs.

In response to the changing epidemic, increased focus has been recommended not only on supply chains for prescribed and illicit opioids, but also on root societal causes of opioid dependence. Rising opioid fatality rates contribute to declining U.S. life expectancy, [3] along with a few other causes of death, commonly referred to as "deaths of despair." [4] This term provides a useful contextual framework for studying socioeconomic risk factors of opioid overdoses and interventions to prevent associated fatalities. Nationwide, the rising rate of fatal opioid overdoses has disproportionately [5] but not exclusively [6] affected Whites, men and middle-aged individuals. The highest opioid overdose death rates are reported in Mountain, Rust Belt, and New England states as well as the South. [7] In 2017, a shift was seen in the urbanicity of the epidemic, with higher opioid overdose death rates in urban than rural areas. [8] At-risk socioeconomic groups for fatal drug use include middle-aged men and women [9], people in lower income strata, insecure housing, those who did not graduate from high school, and recently released prisoners. [10] People who are divorced or separated are also at increased risk for fatal opioid overdose. [11] Although data on SES attributes including education, income, and employment are available at the county [12] and census tract-level [13], the gold standard for analysis is use of individual-level data to examine effects of personal attributes. There is a paucity of individual-level data on prospective relationships between individual-level SES measures and risk of fatal opioid overdose, including for critical factors such as health insurance coverage, employment and marital status, and incarceration. [14]

National surveillance systems for opioid mortality typically do not capture detailed individual-level SES data. [1, 5, 6, 7, 8] Well-designed studies that include these data are often set in smaller geographic areas such as states, [11] and are not generalizable to the U.S. population.

In this study we analyzed individual-level residential, demographic, and SES data from the Mortality Disparities in American Community Study (MDAC). [15] The study included 3,934,000 people whose 2008 American Community Survey (ACS) [13] responses were linked to the National Death Index through 2015 [16] for longitudinal analysis. The MDAC database supported our aim to estimate hazard ratios for demographic, geospatial, and individual-level SES risk factors and fatal opioid overdose in the United States.

## Methods

The MDAC study [15] is a collaborative project of the U.S. Census Bureau; the Centers for Disease Control and Prevention (CDC), National Centers for Health Statistics; the Center for Medicare and Medicaid Services, and the National Institute on Aging and National Heart, Lung, and Blood Institute of the National Institutes of Health. The 2008 wave of the ACS [13] was linked to NDI [16] death certificate records from 2008 to 2015 to create the nationally representative MDAC database of children and adults. Linkage was based on either social security number or the dyad of first and last name, and date of birth. Up to 10 possible matches per ACS record were returned to the Census Bureau. Results were then run through a Census Bureau algorithm that used address, family member names, and other demographic and SES characteristics to assess whether a match was a true match.

The sample frame for the ACS is derived from the Master Address File. Sampling is designed to approximate age, sex, race, Hispanic ethnicity, and state of residence distributions observed in the Census Bureau's annual United States population estimates. Whites were oversampled by about 2% compared to the 2010 Census, and therefore observations for this group were assigned weights of slightly less than one. Other racial/ethnic groups were undersampled by one percent or less, given weights slightly above one. Overall weights were centered on one observation per respondent i.e., the weighted sample approximates the number of 2008 ACS responses rather than the U.S. population. Table cell counts were rounded to four significant digits to prevent disclosure of identity.

The MDAC reference manual [15] describes variables used in this analysis. Minimal data for analysis required cause of death, age, race, Hispanic ethnicity, sex, disability, marital status, employment status, educational attainment, citizenship, housing tenure, rurality, health insurance status, incarceration, household poverty, and Census Division. NDI data indicated if respondents died over 7-year follow-up, from date of ACS 2008 survey to December 31, 2015. International Classification of Diseases, 10th Revision (ICD–10) [17] mortality codes defined cause of death. Since non-specific fatal opioid overdoses account for the majority of fatal opioid overdoses on death certificates [7], opioid overdoses were defined in one category, using ICD–10 multiple-cause-of death codes T40.0-T40.4 and T40.6 (opium, heroin, other opioids, methadone, synthetic narcotics other than methadone, and unspecified narcotic; respectively). Poverty status was defined using Census Bureau methods, based on total family income in the past 12 months, family size, and age composition. If total income of the householder's family was less than the 2008 threshold for the family, the person was considered "below the poverty level," together with every member of his or her family. Matrix tables for poverty level calculations do not vary across the 50 states and the District of Columbia.

The initial MDAC dataset included over 4,512,000 participants who resided in over 73,000 census tracts across 3,000 county equivalents located in the 50 states and the District of Columbia. We excluded people younger than 10 because the small number of opioid fatalities in this age group would require their data to be suppressed to avoid disclosure of identity and 578,000 (12.8%) ACS records without data for NDI linkage (i.e., social security number or name and

date of birth). Comparisons of age, sex, race, and ethnicity of respondents in MDAC and those without data needed for inclusion revealed no overt biases.

Cox Proportional Hazard Models [18] were developed (SAS v9.4, Cary, NC) to estimate hazard ratios (HR) and 95% confidence intervals for variables of interest using weighted, unrounded cell counts. Person-years-contributed were calculated and time-to-event was determined for opioid overdoses and other causes of death. Person-years were censored at time of death for causes other than opioid overdose and on December 31, 2015 for people who were alive (i.e. no NDI record). Multistage sampling of the ACS were conducted using the PHREG procedure (SAS v9.4, Cary, NC).

Referent groups were selected based on a priori data on groups with lowest risk of opioid overdose death, with accommodation to ensure there were sufficient events in the referent group to yield a stable HR. Thus Hispanics were selected as the referent race/ethnicity group rather than Asians and Pacific Islanders and continuous age and age squared variables (to adjust for nonparametric distribution) were used to analyze the effect of age. In addition to univariate models, partially adjusted models included age, sex, and race/ethnicity. The final model included demographic, geographic, and SES. This design enabled evaluation of changes in effects as demographic and SES covariates of interest [19] were added to models. To limit collinearity among variables, educational attainment and employment status were retained while occupation was eliminated. Similarly, citizenship status was retained while place of birth was eliminated. Final model development used a forward stepwise regression process (PROC PHREG, SAS v9.4) to select one variable at a time based on statistical significance of the Chi-squared test of regression model parameter estimates. All 15 variables were retained in the final model, listed in order of selection: disability, marital status, employment, age squared, continuous age, race/ethnicity, sex, educational attainment, [19] citizenship, housing status, rural versus nonrural residence, [20] health insurance, incarceration at time of survey, household poverty, and Census Division [21].

The Office of Management and Budget approved collection and analysis of de-identified ACS data. To meet Title 13 privacy protection requirements, a research proposal was submitted to the MDAC Steering Committee, with data access by a Census analyst assistance, a processes available to all investigators. Output was reviewed by the Census Bureau Disclosure Review Board to ensure confidentiality.

## Results

After exclusion of participants whose ACS responses could not be linked to the NDI, followed by weighting to age, sex, race, Hispanic ethnicity, and state of residence distributions of the U. S. population, and rounding weighted results to four significant digits for privacy protection data on 3,934,000 people were available for analysis. Among them 3800 were classified as having died from an opioid overdose. Another 264,000 died of other causes and 3,666,000 were alive at the end of 2015. A total of 13,620 person-years were contributed by people who died of opioid overdoses, with 1,165,000 person-years among people who died of other causes, and 27,290,000 person-years for people who were classified as alive on December 31, 2015.

Table 1 presents MDAC counts by vital status at end of follow-up. Differentials in cause of death were seen across age groups. People who were 10 to 19 years of age accounted for 8.8% of fatal opioid overdoses and 0.8% of other deaths. People who were 20 to 39 years of age accounted for 41.1% of opioid overdose deaths and 3.7% of other deaths. People in the 40 to 59 years age group accounted for 43.8% of opioid overdose deaths and 17.4% of other deaths. People who were 60 to 79 years of age accounted for 5.4% of opioid-related deaths but 41.0%

**Table 1. Opioid overdose deaths, deaths from other causes, and people alive at end of study follow-up period (2008–2015), MDAC Study.**

| Attribute | | Opioid Overdose | % | Other Deaths | % | Alive | % |
|---|---|---|---|---|---|---|---|
| Age Group (Years) | | | | | | | |
| | 10 to 19 | 335 | 8.8% | 2238 | 0.8% | 630500 | 17.2% |
| | 20 to 39 | 1557 | 41.1% | 9687 | 3.7% | 1218000 | 33.2% |
| | 40 to 59 | 1660 | 43.8% | 45830 | 17.4% | 1216000 | 33.2% |
| | 60 to 79 | 203 | 5.4% | 108300 | 41.0% | 532300 | 14.5% |
| | 80+ | 36 | 0.9% | 97850 | 37.1% | 69270 | 1.9% |
| Race/Ethnicity* | | | | | | | |
| | Hispanic | 277 | 7.3% | 15690 | 5.9% | 542700 | 14.8% |
| | Asian and Pacific Islander | 29 | 0.8% | 7736 | 2.9% | 173700 | 4.7% |
| | American Indian Alaskan Native | 43 | 1.1% | 1570 | 0.6% | 23870 | 0.7% |
| | Black | 311 | 8.2% | 27870 | 10.6% | 441100 | 12.0% |
| | White | 3057 | 80.7% | 209000 | 79.2% | 2435000 | 66.4% |
| | Other | 73 | 1.9% | 2054 | 0.8% | 49720 | 1.4% |
| Sex | | | | | | | |
| | Female | 1499 | 39.5% | 135000 | 51.1% | 1871000 | 51.0% |
| | Male | 2292 | 60.5% | 129000 | 48.9% | 1795000 | 49.0% |
| Disability | | | | | | | |
| | Not Disabled | 2420 | 63.9% | 116500 | 44.1% | 3257000 | 88.8% |
| | Disabled | 1370 | 36.1% | 147400 | 55.9% | 409400 | 11.2% |
| Marital Status | | | | | | | |
| | Married | 1045 | 27.6% | 114300 | 43.3% | 1696000 | 46.3% |
| | Never Married | 1627 | 42.9% | 28310 | 10.7% | 1406000 | 38.3% |
| | Widowed | 146 | 3.9% | 82040 | 31.1% | 142300 | 3.9% |
| | Separated | 165 | 4.4% | 4846 | 1.8% | 72340 | 2.0% |
| | Divorced | 809 | 21.3% | 34460 | 13.1% | 350100 | 9.5% |
| Employment | | | | | | | |
| | Employed | 1520 | 40.1% | 47870 | 18.1% | 2155000 | 58.8% |
| | Unemployed | 367 | 9.7% | 4205 | 1.6% | 144100 | 3.9% |
| | Not in labor force | 1787 | 47.1% | 210900 | 79.9% | 1000000 | 27.3% |
| | Age less than 16 years | 117 | 3.1% | 984 | 0.4% | 366200 | 10.0% |
| Educational Attainment | | | | | | | |
| | Master/ Doctorate | 99 | 2.6% | 15370 | 5.8% | 292000 | 8.0% |
| | Bachelor's Degree | 259 | 6.8% | 24940 | 9.4% | 537100 | 14.7% |
| | Some College/Associate Degree | 1192 | 31.4% | 56370 | 21.4% | 997900 | 27.2% |
| | High School/GED | 1341 | 35.4% | 93150 | 35.3% | 899500 | 24.5% |
| | Less than High School | 900 | 23.7% | 74090 | 28.1% | 939600 | 25.6% |
| Citizenship | | | | | | | |
| | Not a U.S. citizen | 53 | 1.4% | 7006 | 2.7% | 306300 | 8.4% |
| | U.S. Citizen | 3738 | 99.6% | 256900 | 97.3% | 3360000 | 91.6% |
| Housing tenure | | | | | | | |
| | Own with mortgage | 1377 | 36.3% | 72250 | 27.4% | 1881000 | 51.3% |
| | Group quarters | 267 | 7.0% | 25650 | 9.7% | 93960 | 2.6% |
| | Live in house without rent | 79 | 2.1% | 5464 | 2.1% | 55270 | 1.5% |
| | Own, no mortgage | 616 | 16.2% | 104600 | 39.6% | 614200 | 16.8% |
| | Rent | 1452 | 38.3% | 55910 | 21.2% | 1022000 | 27.9% |
| Rural/Nonrural Residence | | | | | | | |
| | Rural | 784 | 20.7% | 61660 | 23.4% | 854000 | 23.3% |

(*Continued*)

**Table 1.** (Continued)

| Attribute | | Opioid Overdose | % | Other Deaths | % | Alive | % |
|---|---|---|---|---|---|---|---|
| | Nonrural | 3007 | 79.3% | 202300 | 76.6% | 2812000 | 76.7% |
| Health Insurance | | | | | | | |
| | Insured | 2686 | 70.9% | 248200 | 94.1% | 3040000 | 82.9% |
| | Uninsured | 1105 | 29.1% | 15700 | 5.9% | 626000 | 17.1% |
| Incarceration* | | | | | | | |
| | Incarcerated | 178 | 4.7% | 1192 | 0.5% | 30660 | 0.8% |
| | Not incarcerated | 3613 | 95.3% | 262760 | 99.5% | 3635310 | 99.2% |
| Household Poverty | | | | | | | |
| | 500%-999% | 604 | 15.9% | 47870 | 18.1% | 1049000 | 28.6% |
| | 300%-499% | 771 | 20.3% | 54350 | 20.6% | 919300 | 25.1% |
| | 100%-299% | 1263 | 33.3% | 105700 | 40.1% | 1184000 | 32.3% |
| | Less than 100% | 931 | 24.6% | 34140 | 12.9% | 428700 | 11.7% |
| | Other* | 223 | 5.9% | 21860 | 8.3% | 84560 | 2.3% |
| Census Division | | | | | | | |
| | West North Central | 214 | 5.6% | 18760 | 7.1% | 242100 | 6.6% |
| | New England | 167 | 4.4% | 12370 | 4.7% | 176100 | 4.8% |
| | East South Central | 246 | 6.5% | 18600 | 7.0% | 214100 | 5.8% |
| | West South Central | 400 | 10.6% | 29230 | 11.1% | 414800 | 11.3% |
| | Pacific | 533 | 14.1% | 35550 | 13.5% | 598700 | 16.3% |
| | Mid Atlantic | 486 | 12.8% | 36200 | 13.7% | 496500 | 13.5% |
| | East North Central | 660 | 17.4% | 42790 | 16.2% | 558300 | 15.2% |
| | South Atlantic | 731 | 19.3% | 53620 | 20.3% | 705900 | 19.3% |
| | Mountain | 352 | 9.3% | 16820 | 6.4% | 259600 | 7.1% |

* Non-Hispanic Race and Hispanic Ethnicity. Incarceration at time of survey

Cell frequencies are weighted to the U.S. population, with rounding according to Census Bureau identity protection rules.

DRB release numbers: CBDRB-FY19-304.

of other deaths, while those and 80+ years of age accounted for 0.9% of opioid-related deaths but 37.1% of other deaths.

Whites accounted for 80.7% of opioid overdose deaths and 66.4% of people who were alive at the end of follow-up. American Indians and Alaskan Natives and Asians and Pacific Islanders each accounted for approximately 1% of opioid overdose deaths, and Blacks and Hispanics accounted for 8.2% and 7.3% of these deaths, respectively. Women accounted for 39.5% while men accounted for 60.5% of fatal opioid overdoses. Disabled people accounted for 36.1% of opioid deaths and 11.2% of people who were alive at the end of the study period. The proportion of opioid overdose deaths that occurred among those who had never married was 42.9%, and people who were divorced accounted for 21.3% of fatal opioid overdoses. People who were not in the labor force accounted for 47.1% and the unemployed accounted for 9.7% of opioid overdose deaths.

People whose highest level of educational attainment was a High School diploma or GED only, with no college accounted for 35.4% of opioid overdose deaths, with 23.7% of opioid overdose deaths among people who did not complete high school. U.S. citizens accounted for 98.6% of opioid overdose deaths. Those who rented accounted for 38.3% of opioid overdose deaths. People residing in nonrural areas accounted for 79.3% of opioid overdose deaths while rural residents accounted for 20.7% of opioid overdose deaths. Those who were uninsured accounted for 29.1% of fatal opioid overdoses. People who were incarcerated at the time of

their survey accounted for 4.7% of fatal opioid overdoses. People living below the poverty line accounted for 24.6% of opioid overdose deaths and 11.7% of those who were alive at the end of the study. The distributions of opioid overdose deaths across Census Division were generally similar to those for other deaths and people alive at the end of the study period.

Hazard ratios and 95% CIs for univariate, partially adjusted models, and a final proportional hazard model are presented in Table 2. In the final model, results included the following: A statistically significant HR and 95% confidence interval (CI) was seen for age, as a continuous variable (HR = 1.18, CI: 1.16, 1.20), reflecting that 85% of deaths occurred among adults 20 to 59 years of age at time of ACS survey. The HR for age squared was 1.00. Compared to Hispanics, several non-Hispanic racial groups had significantly elevated hazard ratios for fatal opioid overdose including Whites (HR = 2.52, CI: 2.21, 2.88); Others (HR = 2.09, CI: 1.61, 2.72); and American Indians and Alaskan Natives (HR = 1.88, CI: 1.35, 2.62). In univariate and partially adjusted models Blacks had increased hazard rates (HR = 1.37, CI: 1.16, 1.60) and (HR = 1.44, CI: 1.22, 1.69), respectively, although those rates became lower than 1 after further adjusting for SES in the final model (HR = 0.81, CI: 0.68, 0.96). A strong protective effect was seen in all models for Asian and Pacific Islanders including the final model (HR = 0.55, CI: 0.37, 0.80). Compared to women, men were at greater risk of fatal opioid overdose (HR = 1.61, CI: 1.50 to 1.72).

Compared to people without disabilities, those who were disabled had a statistically elevated risk of fatal opioid overdose (HR = 2.80, CI: 2.59, 3.03). Compared to married people, those who never married had an elevated HR for opioid-related mortality (1.71, CI: 1.55, 1.89) as did people who were widowed (HR = 2.44, CI: 2.03, 2.95); separated (HR = 2.16, CI: 1.82, 2.56); and divorced (HR = 2.19, CI: 1.99, 2.42). People who were unemployed had higher HRs than those who were employed (HR = 2.46, CI: 2.17, 2.79).

In the final model, compared to people with master or doctoral degrees, those with bachelor degrees only had no statistically significant difference in risk (HR = 1.17, CI: 0.93, 1.47) with statistically significant elevated HRs among those with attainment of a High School diploma or GED only (HR = 2.48, CI: 2.00, 3.06), who did not complete High School (HR = 2.26, CI: 1.81, 2.82), and some college or an Associate Degree (HR = 2.63, CI: 1.84, 2.79). Compared to non-citizens, U.S. citizens had elevated risk (HR = 4.62, CI: 3.48, 6.14). Compared to people who owned a home with a mortgage, HRs for people in other housing were higher including those who owned a home without a mortgage (HR = 1.20, CI: 1.09, 1.33) or rented (HR = 1.36, CI: 1.25, 1.48). Urban residents were more likely than rural residents to die from opioid overdoses (HR = 1.46, CI: 1.34, 1.59). Compared to people with health insurance, those who were uninsured had a significantly higher HR for fatal opioid overdose (HR = 1.30, CI: 1.20, 1.41). Compared to people who were not incarcerated at the time of their 2008 ACS survey, those who were incarcerated had a statistically elevated HR for opioid-related mortality (HR = 2.70, CI: 1.91, 3.81).

Compared to people who lived in households at 500% above the poverty line or more, those living in less affluent households had statistically significantly higher opioid mortality HRs, with the highest HR among people in households below the poverty line (HR = 1.36, CI: 1.20, 1.54). Compared to residents of the West North Central Census division, risk was higher for those in Mountain (HR = 1.58, CI: 1.33, 1.88); South Atlantic (HR = 1.29, CI: 1.11, 1.50); East North Central (HR = 1.27, CI: 1.09, 1.48); Mid Atlantic (HR = 1.25, CI: 1.06, 1.47) and Pacific (HR = 1.19, CI: 1.01, 1.40) Census divisions.

## Discussion

This nationally representative MDAC observational study provides new insights into relationships between SES and opioid-related mortality. A principal finding was that the risk of fatal

**Table 2. Hazard ratios and confidence limits, opioid overdose deaths, 2008–2015, Mortality Disparities in American Communities (MDAC) Study.**

| Attribute | Category | Univariate Models | | Partially Adjusted Models[a] | | Final Model[b] | |
|---|---|---|---|---|---|---|---|
| | | HR | 95% CI | HR | 95% CI | HR | 95% CI |
| Age (Years) | | | | | | | |
| | Continuous | 1.12*** | (1.11, 1.14) | 1.13*** | (1.11, 1.14) | 1.18*** | (1.16, 1.20) |
| Age Squared (Years) | | | | | | | |
| | Continuous | 1.00*** | (1.00, 1.00) | 1.00*** | (1.00, 1.00) | 1.00*** | (1.00, 1.00) |
| Race/Ethnicity† | | | | | | | |
| | Hispanic | Ref | | Ref | | Ref | |
| | Black | 1.37*** | (1.16, 1.60) | 1.44*** | (1.22, 1.69) | 0.81* | (0.68, 0.96) |
| | Asian and Pacific Islander | 0.33*** | (0.23, 0.48) | 0.34*** | (0.23, 0.49) | 0.55** | (0.37, 0.80) |
| | American Indian/Alaskan Native | 3.44*** | (2.50, 4.75) | 3.60*** | (2.61, 4.97) | 1.88*** | (1.35, 2.62) |
| | White | 2.40*** | (2.12, 2.71) | 2.62*** | (2.31, 2.96) | 2.52*** | (2.21, 2.88) |
| | Other | 2.87*** | (2.22, 3.71) | 3.17*** | (2.45, 4.10) | 2.09*** | (1.61, 2.72) |
| Sex | | | | | | | |
| | Female | Ref | | Ref | | Ref | |
| | Male | 1.59*** | (1.49, 1.69) | 1.55*** | (1.45, 1.65) | 1.61*** | (1.5, 1.72) |
| Disability | | | | | | | |
| | Not Disabled | Ref | | Ref | | Ref | |
| | Disabled | 3.99*** | (3.73, 4.26) | 5.48*** | (5.12, 5.87) | 2.80*** | (2.59, 3.03) |
| Marital Status | | | | | | | |
| | Married | Ref | | Ref | | Ref | |
| | Never Married | 1.92*** | (1.78, 2.08) | 2.92*** | (2.66, 3.21) | 1.71*** | (1.55, 1.89) |
| | Widowed | 1.37*** | (1.15, 1.63) | 4.34*** | (3.59, 5.24) | 2.44*** | (2.03, 2.95) |
| | Separated | 3.70*** | (3.14, 4.37) | 4.09*** | (3.47, 4.83) | 2.16*** | (1.82, 2.56) |
| | Divorced | 3.70*** | (3.37, 4.05) | 3.63*** | (3.31, 3.98) | 2.19*** | (1.99, 2.42) |
| Employment | | | | | | | |
| | Employed | Ref | | Ref | | Ref | |
| | Unemployed | 3.61*** | (3.22, 4.04) | 4.19*** | (3.73, 4.70) | 2.46*** | (2.17, 2.79) |
| | Not in labor force | 2.34*** | (2.19, 2.51) | 4.66*** | (4.31, 5.04) | 2.46*** | (2.25, 2.68) |
| | Age less than 16 years | 0.46*** | (0.38, 0.56) | 1.62*** | (1.28, 2.06) | 1.06 | (0.83, 1.34) |
| Educational Attainment | | | | | | | |
| | Master/ Doctorate | Ref | | Ref | | Ref | |
| | Bachelor's Degree | 1.43** | (1.13, 1.80) | 1.37** | (1.09, 1.73) | 1.17 | (0.93, 1.47) |
| | Some College/Associate Degree | 3.51*** | (2.86, 4.30) | 3.71*** | (3.02, 4.56) | 2.26*** | (1.84, 2.79) |
| | High School/GED only | 4.29*** | (3.50, 5.26) | 4.84*** | (3.95, 5.94) | 2.48*** | (2.00, 3.06) |
| | Less than High School | 2.79*** | (2.27, 3.43) | 5.78*** | (4.65, 7.19) | 2.26*** | (1.81, 2.82) |
| Citizenship | | | | | | | |
| | Noncitizen | Ref | | Ref | | Ref | |
| | U.S. Citizen | 6.31*** | (4.81, 8.28) | 4.78*** | (3.61, 6.32) | 4.62*** | (3.48, 6.14) |
| Housing tenure | | | | | | | |
| | Own with mortgage | Ref | | Ref | | Ref | |
| | Group quarters | 3.66*** | (3.21, 4.17) | 4.45*** | (3.87, 5.10) | 0.88 | (0.64, 1.20) |
| | Live in house without rent | 1.90*** | (1.52, 2.39) | 2.34*** | (2.16, 2.53) | 1.21 | (0.96, 1.53) |
| | Own, no mortgage | 1.29*** | (1.17, 1.42) | 1.72*** | (1.56, 1.90) | 1.20*** | (1.09, 1.33) |
| | Rent | 1.93*** | (1.79, 2.08) | 2.14*** | (1.71, 2.69) | 1.36*** | (1.25, 1.48) |
| Urban/Rural Residence | | | | | | | |
| | Rural | Ref | | Ref | | Ref | |
| | Urban | 1.17*** | (1.08, 1.26) | 1.39*** | (1.28, 1.50) | 1.46*** | (1.34, 1.59) |

(*Continued*)

**Table 2.** (Continued)

| Attribute | Category | Univariate Models | | Partially Adjusted Models[a] | | Final Model[b] | |
|---|---|---|---|---|---|---|---|
| | | HR | 95% CI | HR | 95% CI | HR | 95% CI |
| Health Insurance | | | | | | | |
| | Insured | Ref | | Ref | | Ref | |
| | Uninsured | 2.05*** | (1.91, 2.20) | 2.09*** | (1.94, 2.25) | 1.30*** | (1.20, 1.41) |
| Incarceration‡ | | | | | | | |
| | Not incarcerated | Ref | | Ref | | Ref | |
| | Incarcerated | 5.95*** | (5.12, 6.92) | 4.95*** | (4.22, 5.80) | 2.70*** | (1.91, 3.81) |
| Household Poverty | | | | | | | |
| | 500%-999% | Ref | | Ref | | Ref | |
| | 300%-499% | 1.45*** | (1.30, 1.61) | 1.59*** | (1.43, 1.77) | 1.13* | (1.01, 1.26) |
| | 100%-299% | 1.82*** | (1.65, 2.01) | 2.46*** | (2.22, 2.72) | 1.12* | (1.01, 1.25) |
| | Less than 100% | 3.73*** | (3.37, 4.13) | 5.57*** | (5.00, 6.21) | 1.36*** | (1.20, 1.54) |
| | Other | 4.37*** | (3.74, 5.09) | 6.15*** | (5.22, 7.24) | 0.83 | (0.54, 1.29) |
| Census Division | | | | | | | |
| | West North Central | Ref | | Ref | | Ref | |
| | New England | 1.08 | (0.88, 1.32) | 1.12 | (0.92, 1.37) | 1.10 | (0.90, 1.35) |
| | East South Central | 1.30** | (1.08, 1.56) | 1.36** | (1.13, 1.64) | 1.13 | (0.94, 1.36) |
| | West South Central | 1.10 | (0.93, 1.29) | 1.33*** | (1.13, 1.58) | 1.16 | (0.98, 1.37) |
| | Pacific | 1.02 | (0.87, 1.19) | 1.34*** | (1.14, 1.57) | 1.19* | (1.01, 1.40) |
| | Mid Atlantic | 1.11 | (0.95, 1.30) | 1.29** | (1.09, 1.51) | 1.25** | (1.06, 1.47) |
| | East North Central | 1.34*** | (1.15, 1.56) | 1.40*** | (1.2, 1.64) | 1.27** | (1.09, 1.48) |
| | South Atlantic | 1.17* | (1.01, 1.37) | 1.35 | (1.16, 1.58) | 1.29** | (1.11, 1.50) |
| | Mountain | 1.54*** | (1.30, 1.83) | 1.71*** | (1.44, 2.03) | 1.58*** | (1.33, 1.88) |

* P < 0.05

** P < 0.01

*** P<0 .001

† Hispanic ethnicity, non-Hispanic Race.

‡ Incarceration at time of survey

[a]Partially adjusted models adjusted for age variables, non-Hispanic race/Hispanic ethnicity, and sex.

[b]Final model adjusted for age variables, non-Hispanic race/Hispanic ethnicity, sex, disability, marital status, employment, education, citizenship, housing tenure, rural/nonrural, health insurance, incarceration, household poverty, and census division.

Abbreviations: HR: Hazard Ratio, CI: Confidence Interval

Statistics are based on weighted data.

DRB release numbers: CBDRB-FY19-301, CBDRB-FY19-302, CBDRB-FY19-348, CBDRB-FY19-555

opioid overdose was greater among people in low compared to high SES strata. Economic deprivation is a risk factor for opioid overdoses in the United States and contributes to patterns of declining life expectancy that differ from most developed countries. [22] As previously reported, affected demographic groups included adolescent, young and middle-aged adults, [23] Whites, American Indians and Alaskan Natives, people of unspecified race, and men. [24] These findings may be of use in developing targeted efforts to prevent fatal opioid overdoses. [25, 26, 27]

Compared to Hispanics, Whites had the highest HRs for opioid overdose death. This disparity, affecting the largest U.S. racial group, [5] has been attributed to socioeconomic despair [4] and limited opportunity in distressed U.S. communities [28] Other explanations U.S. policy priorities for health care, healthy behaviors, and the physical and social environment.

Other countries with higher life expectancy are outperforming the United States with respect to education, child poverty, and other measures of well-being [9]. American Indians and Alaskan Natives were also at risk for opioid fatality. While Asian and Pacific Islander and Black race were both protective in the most adjusted final model, the latter result should be interpreted with cautiously. In contrast to the protective effect in the final model for Black race that adjusted for 10 SES risk factors, less adjusted models showed significantly elevated HRs of opioid overdose fatality among Blacks. If SES partially mediated the effect of race in the final model, then interventions that impact SES, such as improving education, may have among their many advantages, the ability of help decrease opioid overdoses and associated racial disparities. [25–28] Furthermore, although the opioid epidemic has been most acute in some areas with low percentage Black populations, [7] (e.g., Appalachia, New England, the Midwest, and Mountain states), the recent shift toward a more urban centered opioid overdose epidemic [8] and high opioid overdose fatality rates in other areas with sizeable Black populations (e.g., Southeastern U.S.) could place Blacks at risk moving forward.

People who were disabled had almost three times higher risk of death from opioid overdose than those without a disability, likely reflecting use of opioid analgesics to treat chronic pain. In 2016, CDC published guidelines to assist prescribers in weighing the benefits and risks of opioid therapy for chronic pain. [29] In 2019, the guidelines were evaluated by a consensus panel [30] and the CDC published a perspective [31] on measures to prevent misapplications of the guidelines that can cause harm. Examples include inflexible application of dosage and duration thresholds, abrupt tapering of opioid dosages, drug discontinuation, or dismissal of patients from care. Misapplication of the guidelines to other patient populations is another concern. This includes patients with pain at end-of-life, from cancer, acute surgical recovery, sickle cell crises. Application of chronic pain dosage guidelines when prescribing opioid agonists to treat opioid use disorder can also cause harm. A consensus report highlights national gaps in evidence-based care for opioid use disorder that can save lives. [32] A need exists for empathetic chronic pain management such that non-opioid treatment is provided to the need for opioids, while taking into consideration the risks associated with each type of treatment. When patients agree to taper the dose of opioids, it is helpful for the pace to be individualized and gradual, to minimize withdrawal symptoms. [32] Further research on alternative chronic pain management strategies could point to interventions that lower opioid overdose mortality among patients at risk for opioid use disorder because of their medical comorbidities. [33]

Compared to people who were married, those who were divorced, separated or widowed had higher risk of opioid overdose death, confirming previous associations with fatal opioid overdose. [10, 34] Although the reason for this finding is unclear, behavioral, physical, and economic benefits of having a spouse could confer health benefits. [35] Being in a marital relationship or other domestic partnership may limit time spent alone or social isolation that predisposes to fatal opioid overdose. [35] Research on beneficial effects of interpersonal connection with respect to opioid use disorder could uncover potential interventions to build community resilience to the opioid crisis.

As in previous U.S. studies, people who were unemployed were at greater risk of dying from an opioid overdose compared to the employed [24]. In the U.S., when economic shocks cause rising unemployment, increased risk of opioid-related mortality is seen [36]. In disadvantaged communities, manual labor occupations with higher injury risk are often the most available employment opportunities. Occupational injuries can lead to chronic painful conditions, disability, unemployment and resulting use of opioid analgesics. [37] In one study, family members retrospectively suggested that individual predilection toward unemployment may have contributed to their decedents fatal opioid overdose. [38]. The effectiveness of outreach

efforts could be improved with better understanding of the interconnections between unemployment and risk of fatal opioid overdose [6].

People with less than a four-year college degree had elevated HRs for opioid overdose mortality compared to those with graduate degrees. This is consistent with a previous studies of nonmedical opioid use including nationwide surveys of adolescents and young adults, [39, 40] and studies in smaller areas [10]. This increase in risk of opioid misuse among people with low educational attainment may partially reflect downstream consequences such as less access to stable employment opportunities. [36]

Non-citizens were at lower risk of opioid overdose mortality than citizens. Explanations may include NDI artifacts related to nativity [41], less intensive marketing, or access to opioids among non-citizens. [2] Health affirming values placed on social mobility and family cohesion within traditional [42, 43] or immigrant communities [44] may also mitigate economic stressors that contribute to depression and substance abuse. [42, 43, 44] Cultivating networks of support and resilience within communities affected by the opioid epidemic could help to prevent fatal overdoses in the United States.

Compared to people who owned a house with a mortgage, those who rented were at increased risk of fatal opioid overdose. This finding is consistent with other evidence of health disparities by housing tenure [45–47]. Injected drug use is more frequently reported among people living in unstable housing situations. [48] A study of housing relocation in Atlanta suggest that drug use wains when people move from these settings to neighborhoods with more economic advantage. [49] Authors of the Atlanta study recommended research on barriers that prevent people who use substances from obtaining housing in less disadvantaged neighborhoods. Campaigns to enhance quality of life in less affluent housing neighborhoods may also have merit. An unexpected finding of the present study was that homeowners without mortgages had elevated risk of opioid overdose death compared to those with mortgages. One plausible explanation is that pressure to make scheduled mortgage payments provides routine structure in daily life that discourages opioid misuse.

Based on the Census Bureau definition of rural versus nonrural, in which less than one quarter of the population lived in rural areas, nonrural residents were at 45% greater risk of an opioid fatality than rural residents. This contrasts with other studies in which more rapid increases in prescription opioid mortality rates were reported in rural than nonrural areas early in the opioid epidemic. [50] Higher opioid overdose death rates were also reported in rural states, using a broad definition of rurality. [8] Our finding is however consistent with recent national data [51] showing higher opioid poisoning in urban areas, including from heroin and synthetic opioids, with higher rates of deaths in rural areas from semisynthetic opioids (e.g., oxycodone, hydrocodone, and codeine). A recent study in 17 states indicated that economic disadvantage was a risk factor for prescription opioid overdose death regardless of urbanicity, however economic disadvantage played a larger role in heroin overdose deaths in urban than rural neighborhoods [52]. A national survey found that urban adults were more likely to engage in prescription opioid misuse compared to rural adults. [53] Differences in opioid fatality risk across census divisions in this report were less pronounced than those reported at the state-level [54]. The dispersed geography of opioid overdose deaths in the United States poses an intervention challenge, with variation in rates and trends across jurisdictions influenced by population density, [52] opioids circulating within the community, [53–56] and area-level economic distress. [52]

Risk of death from opioid overdose was associated with not having health insurance. Opioid addiction often occurs amid economic and health problems that can lead to un-insurance [57] Affected U.S. population subgroups are heterogeneous. Tailored responses are therefore needed to deliver appropriate mental health, substance abuse, and social services. [58] Affected

groups include childbearing women and prenatally exposed infants, [59] those at-risk for or with a history of incarceration [60], homeless people, [48] and people living with chronic pain [55, 56]. As responses to the opioid epidemic scale-up to address effects of lack of insurance, training of prescribers can help them to distinguish medical needs from situations in which opioids are likely to be diverted for non-medical use. [24] Provider education can also maximize harm reduction (e.g. naloxone co-prescribing in the context of pain treatment). [32]

Compared to non-incarcerated people, those who were incarcerated were at increased risk of opioid mortality. A need exists for medical and behavioral opioid use disorder therapy during and after incarceration. [60] Lack of access to opioids during incarceration can cause tolerance to diminish, leaving recently incarcerated people susceptible to overdose if they use doses similar to those prior to their incarceration. In Washington State during the 2000s, for example, fatal overdose was the leading cause of death in the 30 days after release. [61] Increased exposure to fentanyl contributes to an emerging pattern of post-incarceration opioid overdose fatality, with longer median time from release fatal overdose. [62] This is not due solely to reduced tolerance but also to increased drug lethality when a previously incarcerated person does encounter (mostly illicit) fentanyl. [62] Quality transition of care for people with opioid use disorders before and after prison release could prevent fatal overdoses in this population. [60]

Consistent with other studies of SES and opioid overdose mortality, [11, 63] compared to people from the most affluent households, those living under the poverty line had higher risk of fatal opioid overdose. Some experts [64] recommend interventions that include treatment of people with opioid use disorder in conjunction with long-term efforts to reduce the opioid supply. [65] Others recommend using a social determinants of health framework to address causes of drug demand, such as loss of opportunity. [2]

This study has strengths that include the nationally representative survey of both children and adults, weighting to adjust for underrepresented groups, the prospective study design, and detailed self-reported SES data. Limitations include potential misclassification of mortality on death certificates, [66] absence of data on psychiatric diagnoses and access to naloxone, and the ascertainment of time-varying measures (e.g., employment, health insurance) only at baseline, up to 7 years before death. Future studies can build on our findings with novel or more detailed SES predictors. In summary, this study provides insights into relationships between SES and U.S. opioid overdose mortality. While opioid fatalities occurred across SES strata, they were concentrated in lower SES groups. These SES attribute specific findings may facilitate the design of opioid overdose prevention, treatment, and rehabilitation programs. [26]

**Access to Data:** The authors of the present study had no special access privileges in accessing MDAC which other interested researchers would not have. In compliance with privacy protection requirements of U.S. Code Title 13, data access was obtained by submitting a research proposal form to the MDAC Steering Committee. [12] Data were accessed with Census Bureau analyst assistance rather than at a Research Data Center, the two processes available to all investigators to access the full MDAC dataset.

**Funding:** MDAC is funded by National Heart, Lung, and Blood Institute and National Institute on Aging agreements with the Census Bureau.

**Census Bureau Disclaimer:** This paper is released to inform interested parties of research and encourage discussion. Views expressed on statistical, methodological, technical, or operational issues are those of the authors and not necessarily the U.S. Census Bureau. Results were reviewed by the Census Bureau's Disclosure Review Board (DRB) to prevent disclosure of confidential information. DRB releases: CBDRB-FY19-301, CBDRB-FY19-302, CBDRB-FY19-304, CBDRB-FY19-348, CBDRB-FY19-555. **NIH Disclaimer:** The views expressed in this manuscript are those of the authors and do not necessarily represent the views of the National

Heart, Lung, and Blood Institute; the National Institute of Mental Health; the National Institute on Drug Abuse; the National Institutes of Health; or the U.S. Department of Health and Human Services.

## Acknowledgments

We thank Sean Coady, Matthew Neiman and Norman Johnson for their efforts to establish MDAC dataset.

## Author Contributions

**Conceptualization:** Sean F. Altekruse, William C. Altekruse.

**Data curation:** Candace M. Cosgrove, William C. Altekruse.

**Formal analysis:** Candace M. Cosgrove, Richard A. Jenkins, Carlos Blanco.

**Funding acquisition:** Sean F. Altekruse.

**Investigation:** Sean F. Altekruse.

**Methodology:** Sean F. Altekruse, Candace M. Cosgrove.

**Project administration:** Sean F. Altekruse.

**Resources:** Candace M. Cosgrove.

**Software:** Candace M. Cosgrove.

**Supervision:** Richard A. Jenkins, Carlos Blanco.

**Validation:** Candace M. Cosgrove, William C. Altekruse.

**Visualization:** Sean F. Altekruse, Candace M. Cosgrove, William C. Altekruse, Carlos Blanco.

**Writing – original draft:** Sean F. Altekruse, Candace M. Cosgrove, William C. Altekruse.

**Writing – review & editing:** Sean F. Altekruse, Candace M. Cosgrove, William C. Altekruse, Richard A. Jenkins, Carlos Blanco.

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
