## [Decision Letter · Decision Letter 0]

1 Aug 2019

PONE-D-19-16947

Socioeconomic risk factors for fatal opioid overdoses in the United States: Findings from the Mortality Disparities in American Communities Study (MDAC)

PLOS ONE

Dear Dr Altekruse,

Thank you for submitting your manuscript to PLOS ONE. After careful consideration, we feel that it has merit but does not fully meet PLOS ONE’s publication criteria as it currently stands. Therefore, we invite you to submit a revised version of the manuscript that addresses the points raised during the review process.

Please fully address the comments provided by the two reviewers. My main concerns are related to the methodological decisions that are not explained or clarified in the text. For example, what was the rationale/approach for variable selection in the adjusted analysis? There is no rationale provided for the base, base + one SES factor, or fully adjusted models. In addition there is no rationale for the parameterization of selected factors, nor the selection of reference categories, leading to some strange results with limited utility (i.e., 80+ for the reference group for age, etc.) Did you consider any effect modification analysis, for example to examine the relationship between covariates of interest (i.e., SES) and geographic region?  Given the national data, there could be further exploration of the data and the relationships examined in the paper. Finally, there is limited information provided in the text regarding the measurement or meaning of the covariates included in the models.  

We would appreciate receiving your revised manuscript by Sep 15 2019 11:59PM. To enhance the reproducibility of your results, we recommend that if applicable you deposit your laboratory protocols in protocols.io, where a protocol can be assigned its own identifier (DOI) such that it can be cited independently in the future. For instructions see: http://journals.plos.org/plosone/s/submission-guidelines#loc-laboratory-protocols

We look forward to receiving your revised manuscript.

Kind regards,

Becky L. Genberg

Academic Editor

PLOS ONE

Journal Requirements:

Reviewers' comments:

Reviewer's Responses to Questions

**Comments to the Author**

1. Is the manuscript technically sound, and do the data support the conclusions?

Reviewer #1: Yes

Reviewer #2: Yes

2. Has the statistical analysis been performed appropriately and rigorously? 

Reviewer #1: Yes

Reviewer #2: Yes

3. Have the authors made all data underlying the findings in their manuscript fully available?

Reviewer #1: Yes

Reviewer #2: Yes

4. Is the manuscript presented in an intelligible fashion and written in standard English?

Reviewer #1: Yes

Reviewer #2: Yes

5. Review Comments to the Author

Reviewer #1: This epidemiological study examines the relationship between multiple indicators of SES and fatal overdose in a nationally representative sample and fills an important gap in the overdose literature. The researchers present data drawn from the American Community Survey, which has been linked to fatal overdose data from the National Death Index. While this is an important topic with implications for targeted overdose prevention and response programs, grounding the research in the large body of existing literature on fatal overdose would strengthen the paper. There are also several methodological issues as well as questions around generalizability that need to be addressed. With minor revisions, this paper could be a timely contribution to the overdose literature.

Abstract

The results section of the abstract should be revised; there appears to be incomplete reporting e.g., urban group had higher risk than the rural group but was not reported, “HRs near two” etc and the wording is convoluted at times. The abstract refers to “final models” when the results only report one final model. I suggest rewriting the results section of the abstract to be more concise and more informative.

Introduction

Page 3: Nationwide, fatal opioid overdoses have disproportionately [2] but not exclusively [3] affected men, and particularly adolescent, young and middle-aged adult white men. -Clarify whether this is referring to fatal opioid overdose prevalence or rates over time and the historical period of relevance.

It would be worth mentioning the reason why overdose has been conceptualized as a death of despair to strengthen the rationale for focusing on SES, and to cite the original paper that developed the theory. The cited reference does not use the phrase and so the intended connection is unclear.

Given that this is a national study, more could be provided in the introduction about the relationship between overdose rates and geography to better contextualize the findings (e.g., urban vs. rural settings), including whether overdoses or low SES are known to be concentrated in certain locales.

Much has been written about the current opioid epidemic. A description of the established risk factors of fatal overdose or at minimum including citations of published reviews (e.g., Dasgupta et al., AJPH; Martins et al., AJPH) would nicely set up the subsequent sections and lay the foundation for the known confounders (measured and unmeasured) that should be considered.

Methods:

Page 4: (12.8%) ACS respondents who were missing information necessary to meet the NCHS minimum criteria required for matching – did the authors conduct secondary analyses to understand the degree of bias due to this missingness e.g., if this group differed on socio-demographics?

Page 5: The Census Bureau definition of rural [14] excluded metropolitan and micropolitan areas, incorporated census-designated places with populations of 2500 or more. All residential locations were classified as rural or urban. – This needs to be unpacked; it is an important factor with implications for targeting resources. It appears that rural vs. non-rural would be a more accurate description based on the current text. Could the authors instead use a more granular classification scheme?

Page 6: Analyses that accounted for multistage sampling in the ACS were conducted using the PHREG procedure – clarify the number of levels included in the final model. Did this method account for correlations between individuals surveyed from the same area of residence (e.g., census division)?

Page 6: incarceration at time of survey – clarify here and in the table whether this is referring to the setting in which the survey was conducted or if the individual had a history of incarceration.

Page 6: Were any confounders considered for inclusion in the regression analysis? Access to naloxone is an example. If not, this should be stated in the limitations.

Results:

Page 8: Many variables as modeled contain extremely small cell sizes (e.g., n=36 among ages 80+; n=29 among Asian Pacific Islanders; n=43 American Indian Alaskan Native etc), which calls into question the validity of these results. Where appropriate, I would suggest re-categorization/collapsing those categories into another category to generate larger cell sizes and/or replacing reference groups containing small cells with new reference groups with larger samples; this will affect the precision of the Cox regression models.

Table 2: Clarify whether the presented data are from the Final adjusted model or the Partially adjusted model

Discussion

Page 22: This nationally representative descriptive study – given that the paper includes data analysis, it is observational rather than descriptive.

Page 22: Given that the term “opioid” is often used synonymously with prescription opioid, I suggest highlighting in the opening paragraph that the analysis also included deaths involving non-medical opioids including heroin and fentanyl.

Page 22: What do the authors mean by “rehabilitate those at-risk for opioid overdose?” Replacing this with less stigmatizing language is recommended.

Page 22: were any subpopulations undersampled? E.g., Asian Pacific Islanders; American Indian Alaskan Native Americans. How valid are the findings around race/ethnicity?

Page 23: How does the poverty measure used account for regional differences in cost of living?

Page 24: Compared to people who owned a house with a mortgage, renters or owners without a mortgage were at risk of fatal opioid overdose during the study period – what was the hypothesis? Was this a surprising finding? I would suggest explaining this finding further.

Page 24: The availability and lethality of fentanyl – I suggest clarifying that it is mostly illicitly manufactured fentanyl.

Page 24: Increased access to opioid agonist therapy for treating opioid use disorder may have merit, but currently adherence rates are low – it is unclear what is meant by “low” and what the possible underlying reasons may be. This statement does not add to the discussion and could be removed.

Limitations: How stable are the SES indicators over time? It would be helpful if the authors could explain which SES variables are most susceptible to temporal change and which remain relatively stable over time.

Limitations: Homelessness is highly prevalent among people who misuse opioids, and many of those who overdose are unstably housed. Same with individuals who are incarcerated at the time of the study. How does the ACS account for homeless and incarcerated individuals? If this is a limitation of the study, please state.

Minor comments: typographic errors should be fixed throughout.

Reviewer #2: 

Abstract:

In the abstract, mentioning the American Community Survey (versus MDAC), is a bit confusing, since MDAC is mentioned in the title. I would suggest being consistent from the title to the abstract for those readers who are doing a brief read of your paper. Also, you could clarify that the SES attributes are based on the time of American Community Survey participation.

Introduction

The introduction to this paper is very brief and the first paragraph is a bit disjointed, briefly covering opioid overdose mortality (the focus of this paper), other deaths of despair (which I’m not sure is needed), and SES disparities. In the second paragraph, the authors circle back to state that there is existing research on demographic and geographic trends in opioid overdoses.

The introduction may better justify the analysis if it is expanded. For example, the authors could structure the introduction to present the existing research on opioid overdose mortality and disparities, highlighting the lack of socioeconomic detail. In the second paragraph, they could introduce and expand the existing information on socioeconomic disparities in life expectancy that may be driven by opioid overdoses and other deaths of despair. They could also further explain some of the existing research on distressed communities to motivate looking more into SES indicators.

Methods

What are the NCHS minimum criteria for matching to the NDI you mention in line 84 (i.e. same as those in line 93-94)? Could you add these or provide a reference? Also, define NCHS at first mention in line 84 (later you spell it out without specifying that this is NCHS is lines 92-93) and it makes it seem like there are multiple criteria/procedures.

This sentence needs clarification, I am not sure what you mean by ‘centered on one’: “Frequencies were estimated using weighted data adjusted to be centered on one, with rounding to four significant digits to protect against disclosure of individual information.”

Results

Table 1 has a footnote that says the cell frequencies are based on weighted data, but the frequencies add to the number that I understood to be the unweighted/sampled n ~3.9 million.

The clarity of Table 2 might be improved by presenting the first column as the ‘Base Or Partially Adjusted Model’ – it is a bit confusing to only call that column the Base model and include HRs for SES covariates.

The results section describing the HRs is quite long and needs proofreading. I would suggest condensing this section and only including the key findings to make sure they are not overlooked. For example, you begin the results section describing Table 2/regression models by saying “HRs for many age groups were higher in final than base models (Table 2).” I’m not sure the point of highlighting this since the interest is in the SES variables. Please also be consistent with how the HRs are presented in the results section – it is a bit distracting to see the results presented so many different ways (e.g. mix of HRs inside parentheses or just placed at end of the sentence).

Discussion

In general, I think the discussion needs more focus on the public health impact and potential actions/interventions from the results since you say in the abstract that these results are useful for overdose prevention targeting. I think there are also some opportunities to discuss harm reduction (e.g. naloxone co-prescribing in the context of pain treatment) that are not mentioned.

The paragraph on health insurance is somewhat unclear - I’m not sure how the final two sentences connect to the results and the potential public health implications of this result aren’t clear. For example, should we increase insurance coverage and/or find ways to engage those without health insurance with harm reduction and addiction treatment services?

One potential explanation for the higher overdose mortality among separated, divorced, or widowed people are that they may be more likely to use alone, conferring a higher fatal overdose risk.

I’m a bit confused about how the first and second parts from this sentence are connected: “The availability and lethality of fentanyl has contributed to an emerging pattern of post-incarceration overdose deaths, with a median time from release to death of more than 90 days [40].”

6. PLOS authors have the option to publish the peer review history of their article (what does this mean?). If published, this will include your full peer review and any attached files.

Reviewer #1: Yes: Ju Nyeong Park

Reviewer #2: No

---

## [Author Response · Author response to Decision Letter 0]

2 Oct 2019

Response to Reviewers’ comments: PONE-D-19-16947 

S Altekruse, C Cosgrove, W Altekruse, R Jenkins, C Blanco. Socioeconomic risk factors for fatal opioid overdoses in the United States: Findings from the Mortality Disparities in American Communities Study 

Editor's Query on Data Availability

Editor's Comment 1: Please clarify all data that constitutes minimal data.

Response: On line 101-104 of the manuscript, the authors indicate that the minimal data for the present MDAC analysis were comprised of the following variables: cause of death, age, race, Hispanic ethnicity, sex, disability, marital status, employment status, educational attainment, citizenship, housing tenure, rurality, health insurance status, incarceration, household poverty, and Census Division.

Editor's Comment 2: Please fully disclose the methods of data collection from MDAC that independent researchers can replicate.

Response: As indicated on line 150-152 of the manuscript, in accordance with U.S. Code Title 13 privacy protection requirements, MDAC data were obtained by submitting a request to the MDAC Steering Committee as outlined on the MDAC website: www.census.gov/mdac. Need to know justifications for access to MDAC data include analysis for publication, validation of prior publication findings, and extension of previous research. There are two principal methods available to access the full MDAC dataset. As indicated on line 151-152, the authors of the present study accessed the MDAC dataset in coordination with a Census Bureau analyst. As described on line 359, the other principal MDAC data access option is to independently access the data at a Federal Statistical Research Data Center (FSRDC). FSRDC information is found at the website: www.census.gov/fsrdc. 

Editor's Comment 3: Please confirm the authors of the present study had no special access privileges in accessing MDAC which other interested researchers would not have.

Response: On line 355, the authors of the present study state that they had no special access privileges in accessing MDAC which other interested researchers would not have. 

Academic Editor

Editor Comment 1: What was the rationale/approach for variable selection in the adjusted analysis? 

Response: We thank the editor for this suggestion, which greatly improved the quality of the analysis. On page 7, line 131-134 the method for selecting referent groups is provided. On page 7, line 134-135, the rationale for presenting a partially adjusted model with key demographic variables and a single SES variable are described. On page 7, line 139-141, the approach to retention or elimination of variables when known collinearity exists is described. The forward stepwise model selection process used in the final model is described on page 7, lines 141-149.

“Referent groups were selected based on a priori information on groups with low risk of opioid overdose death, with consideration to ensure sufficient numbers of events occurred in the referent group to estimate stable hazard ratios... To limit effects of collinear variables, educational attainment and employment status were retained while occupation was eliminated from analysis. Similarly, citizenship status was retained while place of birth was eliminated For the purpose of final model development, we started with no predictor variables and used a forward stepwise regression process (PROC PHREG, SAS v9.4) to select one variable at a time based on the statistical significance of the Chi-squared test of the regression model parameter estimates. All 15 candidate variables of interest were retained in the final model, listed here in the order of their stepwise forward selection into the model: disability status, marital status, employment status, age squared (to adjust for nonparametric distribution), continuous age, race/ethnicity, sex, educational attainment, [17] citizenship, employment status, housing status, rural versus nonrural residence, [18] health insurance status, incarceration status at time of survey, household poverty level, and location of residence across nine Census Bureau Divisions of the United States [19].”

Editor Comment 2: There is no rationale provided for base, base+one SES factor and final models. 

Response: On page 7, line 135-139, rationale is provided for univariate models, partially adjusted models included key demographic attributes (age, sex, race/ethnicity) and a single SES or geographic variable.

“In addition to univariate models, partially adjusted models included key demographic attributes (age, sex, race/ethnicity) and a single SES or geographic variable, as well as a final model that included all demographic as well as SES attributes. This enabled evaluation of effects as multiple demographic and SES covariates of interest were added to proportional hazard models. [16]”

Editor Comment 3: There is no rationale for the parameterization of factors, nor selection of reference categories, leading to some strange results of limited utility (i.e., 80+ for the reference group for age). 

Response: In response to this helpful comment, referent groups for age and race/ethnicity were changed in the revised manuscript. On page 7, line 133-136, we describe changing the age variable from an age group category variable to a continuous variable with an age squared term to address potential non-parametric distribution of age and eliminate the small count 80+ group as a referent group. Similarly, with respect to race/ethnicity, the referent group was changed from Asian and Pacific Islander to Hispanic, providing more stable counts for Cox Proportional Hazard Analyses. This suggested revision improve the performance of final models with respect to these variables.

“Referent groups were selected based on a priori information on groups with low risk of opioid overdose death, with consideration to ensure sufficient numbers of events occurred in the referent group to estimate stable hazard ratios. Thus Hispanics were the selected referent race/ethnicity group and continuous age and age squared variables (to adjust for nonparametric distribution) were used to analyze the effect of age.”

Editor Comment 4: Did you consider any effect modification analysis, for example to examine the relationship between covariates of interest (i.e., SES) and geographic region? 

Response: We did not consider interaction terms in this analysis. The goals of this analysis were broad in nature. On page 27, line 348-349, as a limitation we note that many other MDAC analyses of opioid-related mortality are of interest. “In addition, to focus our analyses, we limited our analyses to SES-related variables. Future studies should build on our findings to examine additional predictors of opioid overdose death.”

Editor Comment 5: Given the national data, there could be further exploration of the data and the relationships examined in the paper. 

Response: As indicated in response to the Editor’s comment 4 above we agree with this suggestion and note that this was an overview paper. Our goal is to make the MDAC database known as a resource to explore opioid fatality topics in depth in future papers. 

Editor Comment 6: Finally, there is limited information provided in the text regarding the measurement or meaning of the covariates included in the models. 

Response: In response to this suggestion, on page 5, starting on line 101, the ICD-10 variable for opioid deaths is more fully described. On line 109, readers are referred to the MDAC reference manual for detailed information on each variable used in the analysis. The household poverty variable is now described in detail starting on line 110. Construction of age variables is described on line 133, rural versus non-rural residence is described with reference on line 147. It is clarified that incarceration status was measured at time of survey on line 148. A reference to Census Divisions is provided on line 149. 

Reviewer 1

Reviewer 1 Comment 1: Grounding the research in the large body of existing literature on fatal overdose would strengthen the paper. 

Response: The manuscript refence section was expanded by 10 references, primarily on this topic, including seminal AJPH review articles by Dasgupta et al on socioeconomic status and the opioid epidemic and Martins et al on socioeconomic status of unintentional drug overdose. 

Abstract

Reviewer 1 Comment 2: The results section of the abstract should be revised; there appears to be incomplete reporting e.g., urban group had higher risk than the rural group but was not reported, “HRs near two” etc. and the wording is convoluted at times. 

Response: We thank the reviewer for the suggestion, incorporated throughout the revised manuscript.

Reviewer 1 Comment 3: The abstract refers to “final models” when the results only report one final model. 

Response: This typographical error was corrected.

Reviewer 1 Comment 4: I suggest rewriting the results section of the abstract to be more concise and more informative.

Response: This suggestion was adopted with careful attention to convey the most salient information.

Introduction

Reviewer 1 Comment 5: Page 3: Nationwide, fatal opioid overdoses have disproportionately [2] but not exclusively [3] affected men, and particularly adolescent, young and middle-aged adult white men. -Clarify whether this is referring to fatal opioid overdose prevalence or rates over time and the historical period of relevance. 

Response: On line 63, it is now stated that, “Nationwide, the rising rate of fatal opioid overdoses has disproportionately affected…”

Reviewer 1 Comment 6: It would be worth mentioning the reason why overdose has been conceptualized as a death of despair to strengthen the rationale for focusing on SES, and to cite the original paper that developed the theory. The cited reference does not use the phrase and so the intended connection is unclear. 

Response: On line 61, it is now stated, “This term provides a useful contextual framework for studying SES-related risk factors for fatal opioid overdose and interventions to treat and prevent them.” As recommended by the reviewer, we now cite reference 4, Case and Deaton’s original publication using the term “death of despair.”

Reviewer 1 Comment 7: Given that this is a national study, more could be provided in the introduction about the relationship between overdose rates and geography to better contextualize the findings (e.g., urban vs. rural settings), including whether overdoses or low SES are known to be concentrated in certain locales. 

Response: Continuing on lines 64-65, the introduction includes the following statement with references, 

“Highest opioid death rates are reported in Mountain, Rust Belt, New England states and the South. [7] In 2017, a shift was seen in urbanicity of the opioid epidemic, with higher reported death rates in urban than rural areas. [8] At-risk socioeconomic groups for fatal drug use include people in lower income strata, who did not graduate from high school, the homeless, and recently released prisoners. [9]”

Reviewer 1 Comment 8: Much has been written about the current opioid epidemic. A description of the established risk factors of fatal overdose or at minimum including citations of published reviews (e.g., Dasgupta et al., AJPH; Martins et al., AJPH) would nicely set up the subsequent sections and lay the foundation for the known confounders (measured and unmeasured) that should be considered.

Response: As indicated in response to Reviewer 1, Comment 1 above, the manuscript refence section was expanded by 10 references, primarily on the topic of SES, including the recommended seminal AJPH review articles by Dasgupta et al on socioeconomic status and the opioid epidemic and Martins et al on socioeconomic status of unintentional drug overdose. 

Methods:

Reviewer 1 Comment 9: Page 4: (12.8%) ACS respondents who were missing information necessary to meet the NCHS minimum criteria required for matching – did the authors conduct secondary analyses to understand the degree of bias due to this missingness e.g., if this group differed on socio-demographics? 

Response: On line 120-123, we state, “Census Bureau evaluation of the age, race/ethnicity, and gender of MDAC respondents who were included in MDAC and those without sufficient information to complete an ACS/NDI linkage revealed no biases with respect to these variables [personal communication, Norman Johnson].” ” 

Reviewer 1 Comment 10: Page 5: The Census Bureau definition of rural [14] excluded metropolitan and micropolitan areas, incorporated census-designated places with populations of 2500 or more. Residents were classified as rural or urban. It is an important factor with implications for targeting resources. Rural vs. nonrural is a more accurate label. Could the authors use more granular classification?

Response: We thank the reviewer for this comment. On line 147, the variable is renamed “rural vs. nonrural.” A reference defining the variable is provided. On line 289 we indicate that people residing in nonrural areas accounted for 79% of opioid deaths while rural residents accounted for 21% of opioid deaths. This finding is discussed in the context of the literature in the paragraph beginning on line 289. While the MDAC database does not include a more granular variable for residential density, this variable does appears relevant in the context of the recent literature showing a shift of the plurality of opioid fatalities away from rural settings.

Reviewer 1 Comment 11: Page 6: Analyses that accounted for multistage sampling in the ACS were conducted using the PHREG procedure – clarify the number of levels included in the final model. Did this method account for correlations between individuals surveyed from the same area of residence (e.g., census division)?

Response: The multistage variables used in sample weight assignment are described on line 90-93. 

“The complex sampling frame for the ACS is derived from the Master Address File. Sampling is designed to approximate characteristics observed in the Census Bureau’s annual United States population estimates for the following attributes: age, sex, race, Hispanic ethnicity, and state of residence. Analyses did not account for correlations between individuals within each level for these variables.”

Reviewer 1 Comment 12: Page 6: incarceration at time of survey – clarify here and in the table whether this is referring to the setting in which the survey was conducted or if the individual had a history of incarceration.

Response: On line 148 and in tables it is stated that incarceration status was measured at time of survey. 

Reviewer 1 Comment 13: Page 6: Were any confounders considered for inclusion in the regression analysis? Access to naloxone is an example. If not, this should be stated in the limitations.

Response: On line 342, it is stated that data on access to naloxone was not available.

Results:

Reviewer 1 Comment 14: Page 8: Many variables as modeled contain extremely small cell sizes (e.g., n=36 among ages 80+; n=29 among Asian Pacific Islanders; n=43 American Indian Alaskan Native etc), which calls into question the validity of these results. Where appropriate, I would suggest re-categorization/collapsing those categories into another category to generate larger cell sizes and/or replacing reference groups containing small cells with new reference groups with larger samples; this will affect the precision of the Cox regression models. 

Response: We appreciate this suggestion, and referent groups for age and race/ethnicity were changed in the revised manuscript. Additional information is provided in response to Editor Comment 1.

Reviewer 1 Comment 15: Table 2: Clarify whether the presented data are from the Final adjusted model or the Partially adjusted model

Response: Thank you for pointing this out. On line 191 it is now stated, “In the final model, results for fatal opioid overdose included the following:”

Discussion

Reviewer 1 Comment 16: Page 22: This nationally representative descriptive study – given that the paper includes data analysis, it is observational rather than descriptive. 

Response: In response to this helpful comment, line 235 was revised as follows: “The nationally representative MDAC observational study provides insights into longitudinal relationships between SES and opioid-related mortality.”

Reviewer 1 Comment 17: Page 22: Given that the term “opioid” is often used synonymously with prescription opioid, I suggest highlighting in the opening paragraph that the analysis also included deaths involving non-medical opioids including heroin and fentanyl.

Response: Line 237 states opioids include medical opioids, heroin, fentanyl and its synthetic analogues.

Reviewer 1 Comment 18: Page 22: What do the authors mean by “rehabilitate those at-risk for opioid overdose?” Replacing this with less stigmatizing language is recommended.

Response: Line 243-244 has been revised to be less stigmatizing: “These findings may be of use in developing more targeted efforts to prevent fatal opioid overdoses.”

Reviewer 1 Comment 19: Page 22: were any subpopulations undersampled? E.g., Asian Pacific Islanders; American Indian Alaskan Native Americans. How valid are the findings around race/ethnicity? 

Response: Line 95-97 was revised as follows: “ Whites were oversampled by about 2% compared to the 2010 Census, and therefore this group was assigned a weight of slightly less than one. Other racial/ethnic groups were undersampled by one percent or less and thus given weights above one.”

Reviewer 1 Comment 20: Page 23: How does the poverty measure used account for regional cost of living differences?

Response: This is addressed beginning on line 110: “Derivation of the poverty status variable is described in detail because more than one method is in common use. The variable we used is the standard Census Bureau procedure, involving total family income in the past 12 months, family size and age composition. If the total income of the householder's family was less than the 2008 threshold appropriate for that family, then the person was considered “below the poverty level,” together with every member of his or her family. Matrix tables for poverty level calculations do not vary across the 50 states and D.C.”

Reviewer 1 Comment 21: Page 24: Compared to people who owned a house with a mortgage, renters or owners without a mortgage were at risk of fatal opioid overdose during the study period – what was the hypothesis? Was this a surprising finding? I would suggest explaining this finding further.

Response: Line 285-288 addresses this helpful reviewer’s comment: “An unexpected finding in the present study was that homeowners with mortgages were at elevated risk of opioid death compared to those with a mortgage. A plausible explanation is that the pressure to make scheduled mortgage payments could impose a routine structure into daily life that discourages opioid misuse.”

Reviewer 1 Comment 22: Page 24: The availability and lethality of fentanyl – I suggest clarifying that it is mostly illicitly manufactured fentanyl. 

Response: This suggestion is appreciated. Beginning on line 324 the following revision was made: “These later onset post-incarceration deaths were not thought to be from primarily overdoses due to reduced tolerance but the lethality of mostly illicit fentanyl when encountered.” 

Reviewer 1 Comment 23: Page 24: Increased access to opioid agonist therapy for treating opioid use disorder may have merit, but currently adherence rates are low – it is unclear what is meant by “low” and what the possible underlying reasons may be. This statement does not add to the discussion and could be removed.

Response: The statement beginning line 326 is replaced with, “Ongoing efforts to improve transition of care for people with psychiatric disorders in the months around leaving prison could help to decrease the number of overdoses.”

Reviewer 1 Comment 24: Limitations: How stable are the SES indicators over time? It would be helpful if the authors could explain which SES variables are most susceptible to temporal change and which remain relatively stable over time. 

Response: It would be difficult to rank the SES variables used in this analysis based on the probability of change overtime. On line 343, we attempt to address this limitation as follows: “Furthermore, time-varying measures that can change over time (e.g., employment, health insurance) were assessed once, up to 7 years before death.” 

Reviewer 1 Comment 25: Limitations: Homelessness is highly prevalent among people who misuse opioids, and many of those who overdose are unstably housed. 

Response: The ACS does not ask about the stability of housing. Beginning on line 284, we discuss how the ACS data on housing tenure is consistent with other data on drug use by housing stability. We also cite lack of detail on housing stability as a limitation on line 344: “…we limited our analyses to SES-related variables, some of which lacked optimal detail (e.g., homelessness with housing tenure and incarceration status at time of death).” 

Reviewer 1 Comment 26: Same with individuals who are incarcerated at the time of the study. How does the ACS account for homeless and incarcerated individuals? If this is a limitation of the study, please state.

Response: In the methods, results and discussion we state that incarceration was determined at the time of the 2008 ACS survey. As indicated in response to Reviewer 1, Comment 25, limited detail on incarceration status is listed as a limitation.

Reviewer 1 Comment 27: Minor comments: typographic errors should be fixed throughout.

Response: We have proof read the manuscript in an effort to eliminate typographical errors.

Reviewer #2: 

Abstract:

Reviewer 2 Comment 1: In the abstract, mentioning the American Community Survey (versus MDAC), is a bit confusing, since MDAC is mentioned in the title. I would suggest being consistent from the title to the abstract for those readers who are doing a brief read of your paper. Also, you could clarify that the SES attributes are based on the time of American Community Survey participation.

Response: Line 26-27 were revised as follows: “The Mortality Disparities in American Community Study (MDAC) links population-weighted 2008 American Community Survey responses (n=3,934,000) to the National Death Index through 2015.”

Introduction

Reviewer 2 Comment 2: The introduction to this paper is very brief and the first paragraph is a bit disjointed, briefly covering opioid overdose mortality (the focus of this paper), other deaths of despair (which I’m not sure is needed), and SES disparities. In the second paragraph, the authors circle back to state that there is existing research on demographic and geographic trends in opioid overdoses. The introduction may better justify the analysis if it is expanded. For example, the authors could structure the introduction to present the existing research on opioid overdose mortality and disparities, highlighting the lack of socioeconomic detail. In the second paragraph, they could introduce and expand the existing information on socioeconomic disparities in life expectancy that may be driven by opioid overdoses and other deaths of despair. They could also further explain some of the existing research on distressed communities to motivate looking more into SES indicators.

Response: We appreciate this constructive comment. It was informative in our effort to revise the introduction. The number of citations in the paper was expanded to reference review articles on relationships between SES and drug overdose by Martins et al and Dasgupta et al. The work of Case and Deaton on deaths of despair was also cited, because it provides a framework for examining the relationship between SES indicators and fatal opioid overdoses. 

Methods

Reviewer 2 Comment 3: What are the NCHS minimum criteria for matching to the NDI you mention in line 84 (i.e. same as those in line 93-94)? Could you add these or provide a reference? Also, define NCHS at first mention in line 84 (later you spell it out without specifying that this is NCHS is lines 92-93) and it makes it seem like there are multiple criteria/procedures.

Response: This sentence was revised and is now found on line 119: “The 578,000 (12.8%) ACS records without data for NDI linkage were dropped from the study (i.e., social security number or name and date of birth).” 

Reviewer 2 Comment 4: Please clarify what you mean by ‘centered on one’: “Frequencies were estimated using weighted data adjusted to be centered on one, with rounding to four significant digits to protect against disclosure of individual information.”

Response: Lines 98-100 are revised to indicate weights are centered on one observation per response:

“Overall weights were centered on one observation per respondent, so that the weighted sample approximates the number of 2008 ACS responses rather than the U.S. population.’

Results

Reviewer 2 Comment 5: Table 1 has a footnote that says the cell frequencies are based on weighted data, but the frequencies add to the number that I understood to be the unweighted/sampled n ~3.9 million.

Response: Please see the response to Reviewer 2, Comment 4.

Reviewer 2 Comment 6: The clarity of Table 2 might be improved by presenting the first column as the ‘Base Or Partially Adjusted Model’ – it is a bit confusing to only call that column the Base model and include HRs for SES covariates.

Response: We appreciate this suggestion. Table two now labels the three sets of models as the univariate models, partially adjusted models, and final model. The footnote for the partially adjusted model defines the variables in the models (i.e., age variables, non-Hispanic race/Hispanic ethnicity, and sex alone or with one SES or geographic variable).

Reviewer 2 Comment 7: The results section describing the HRs is quite long and needs proofreading. I would suggest condensing this section and only including the key findings to make sure they are not overlooked. For example, you begin the results section describing Table 2/regression models by saying “HRs for many age groups were higher in final than base models (Table 2).” I’m not sure the point of highlighting this since the interest is in the SES variables. 

Response: The authors appreciate this comment and have incorporated it into the revised results section, by not highlighting the effects for the age (demographic) variable in various models. 

Reviewer 2 Comment 8: Please also be consistent with how the HRs are presented in the results section – it is a bit distracting to see the results presented so many different ways (e.g. mix of HRs inside parentheses or just placed at end of the sentence).

Response: One convention is now used throughout the results section, “(HR=N.NN, CI: N.NN, N.NN)

Discussion

Reviewer 2 Comment 9: In general, I think the discussion needs more focus on the public health impact and potential actions/interventions from the results since you say in the abstract that these results are useful for overdose prevention targeting. 

Response: We appreciate this recommendation and have responded by attempting to expand on potential actions/interventions relevant to SES variables in the discussion.

Reviewer 2 Comment 10: I think there are also some opportunities to discuss harm reduction (e.g. naloxone co-prescribing in the context of pain treatment) that are not mentioned.

Response: Line 313 to 314 states “This effort may require clinician education to distinguish between opioid use disorder and physical dependence to opioids, [32] while maximizing harm reduction efforts (e.g. naloxone co-prescribing in the context of pain treatment).”

Reviewer 2 Comment 11: The paragraph on health insurance is somewhat unclear - I’m not sure how the final two sentences connect to the results and the potential public health implications of this result aren’t clear. For example, should we increase insurance coverage and/or find ways to engage those without health insurance with harm reduction and addiction treatment services?

Response: Beginning on line 308, the paragraph has been revised to make clearer connections between expanded access to care and the role of providers.

“In responses to the national opioid epidemic, individuals affected by addiction who are uninsured may be more able to access critical healthcare resources now than in the past, including mental health and substance abuse services. [30] As healthcare services become more accessible to people at-risk of opioid-related fatality, heightened attention has also been given to responsibilities of prescribers to distinguish between legitimate prescriptions for pain management and the prescription of opioids that could be diverted for non-medical purposes. [31]” 

Reviewer 2 Comment 12: One explanation for the higher overdose mortality among separated, divorced, or widowed people are that they may be more likely to use alone, conferring a higher fatal overdose risk.

Response: On line 255 this sentence was added, “Being a member of a marital relationship or other domestic partnership may limit time spent alone, and resulting social isolation that may predispose to fatal opioid overdose. [29]”

Reviewer 2 Comment 13: How are the first and second parts from this sentence are connected: “The availability and lethality of fentanyl has contributed to an emerging pattern of post-incarceration overdose deaths, with a median time from release to death of more than 90 days [40].”

Response: On line 324, this sentence was revised to, “These later onset post-incarceration deaths were not thought to be primarily due to reduced tolerance but a consequence of the lethality of (mostly illicit) fentanyl when eventually encountered. [52]”

---

## [Decision Letter · Decision Letter 1]

21 Nov 2019

PONE-D-19-16947R1

Socioeconomic risk factors for fatal opioid overdoses in the United States: Findings from the Mortality Disparities in American Communities Study (MDAC)

PLOS ONE

Dear Dr Altekruse,

Thank you for submitting your manuscript to PLOS ONE. After careful consideration, we feel that it has merit but does not fully meet PLOS ONE’s publication criteria as it currently stands. Therefore, we invite you to submit a revised version of the manuscript that addresses the points raised during the review process.

As you revise your manuscript according to the minor revisions suggested by the two reviewers, I also encourage you to give some additional thought to the relationship between race, SES and fatal overdose, and as reviewer #2 has suggested, consider presenting the partially adjusted results in light of the complex relationships between SES and race. 

We would appreciate receiving your revised manuscript by Jan 05 2020 11:59PM. To enhance the reproducibility of your results, we recommend that if applicable you deposit your laboratory protocols in protocols.io, where a protocol can be assigned its own identifier (DOI) such that it can be cited independently in the future. For instructions see: http://journals.plos.org/plosone/s/submission-guidelines#loc-laboratory-protocols

We look forward to receiving your revised manuscript.

Kind regards,

Becky L. Genberg

Academic Editor

PLOS ONE

Reviewers' comments:

Reviewer's Responses to Questions

**Comments to the Author**

1. If the authors have adequately addressed your comments raised in a previous round of review and you feel that this manuscript is now acceptable for publication, you may indicate that here to bypass the “Comments to the Author” section, enter your conflict of interest statement in the “Confidential to Editor” section, and submit your "Accept" recommendation.

Reviewer #1: All comments have been addressed

Reviewer #2: (No Response)

2. Is the manuscript technically sound, and do the data support the conclusions?

Reviewer #1: Yes

Reviewer #2: Yes

3. Has the statistical analysis been performed appropriately and rigorously? 

Reviewer #1: Yes

Reviewer #2: Yes

4. Have the authors made all data underlying the findings in their manuscript fully available?

Reviewer #1: Yes

Reviewer #2: Yes

5. Is the manuscript presented in an intelligible fashion and written in standard English?

Reviewer #1: Yes

Reviewer #2: Yes

6. Review Comments to the Author

Reviewer #1: Major comments

The manuscript has been greatly improved from the previous version. I have a couple of remaining comments.

Abstract: Refrain from using stigmatizing language. Replace “disabled people” with “people with disabilities”

Introduction: The authors have done a nice job of summarizing the literature on the opioid overdose epidemic. An explanation of the gaps and limitations in the literature would round out this section so that the reader can understand the novelty of the current study.

Results: Given the large sample size, I suggest reporting one decimal place for all percentages, both in the text and the tables.

In table 2, level of statistical significance would likely be of interest to readers. I would indicate using footnotes (e.g., * p <0.05, ** p<0.001 etc)

Discussion: Another limitation is that medication and behavior therapy coverage was not measured. The authors may want to mention this fantastic consensus report that highlights national gaps in coverage https://www.nap.edu/catalog/25310/medications-for-opioid-use-disorder-save-lives

Minor comments

Abstract:

Replace “individual” with “individual-level” in the second sentence of the abstract.

In the results, “those with the highest level of educational attainment of high school” does this include college? This finding, as written, does not standalone.

Reviewer #2: Thank you for the opportunity to review this revised manuscript. I have a few additional suggestions to improve the clarity and public health importance of the manuscript.

1. Throughout: use person-first language where you are able. For example, line 68: “the homeless, and recently released prisoners” could be changed to persons who have experienced homelessness, housing instability, or incarceration. Also line 285 “substance users” should be changed to persons who use substances

2. Line 24: “Opioid fatalities are a U.S. problem.” – I think the prior version’s topic sentence was stronger and you might remove this sentence in favor of expanding the second sentence’s justification for the analysis.

3. Line 29: Likewise “HRs for fatal opioid overdose by SES in final model:” in the results section is not needed. Clearer to go straight to the results you interpret.

4. Line 41: “West North Central state”, do you mean “West, North, or Central”? Potentially missing a word or were these areas combined?

5. Lines 45-46: Could strengthen the conclusions by suggesting how these findings from MDAC inform prevention, treatment, and rehabilitation (e.g. what communities or regions need more services?)

6. Introduction lines 69-71: “Some risk factors such as unemployment and health insurance status are optimally studied using individual-level longitudinal data, with adjustment for other covariates. [11]” I do not get a lot of information from this sentence about what is known about the relationship of employment and health insurance coverage with opioid overdose. Could you instead highlight the main finding(s) of citation 11?

7. Introduction, lines 77-78: “after multivariable adjustment” – It may be clearer to say you hypothesized these would remain independently associated with fatal opioid overdose even after adjustment for other socioeconomic and demographic indicators (or something similar). ‘Multivariable adjustment’ on its own leaves the reader unclear on what you are doing.

8. Methods, lines 93-95: “Representativeness of the sample relative to the United States population was enhanced by applying weights to account for variable sampling.” – I don’t think you need this sentence and it makes the description of sampling weights less clear. The phrase ‘variable sampling’ is vague – unclear if variable is used as an adjective (as in varying) or noun (as in age, sex, etc. variables used for sampling).

9. Results, line 156: “weighting to the U.S. population” could be clearer if you say “weighting to the age, sex, etc…distribution of the U.S. population.” Otherwise wording could imply that the sample size is weighted to the size of the US population and someone reading quickly or without much background in survey data analysis might misunderstand that only 3800 opioid overdoses occurred in the US if not clarified that the exact numbers do not represent the US population size.

10. The result in the final model that black participants were at lower risk of fatal overdose should be considered more thoroughly. The unadjusted and partially adjusted models suggest black race is a risk factor for fatal overdose and this association only becomes protective after adjusting for many SES related factors that disproportionately affect African American communities. A model considering race as a main exposure should not be adjusted for SES factors because race would predict some of those SES indicators but the SES factors would be mediators of relationships between race and opioid fatality (i.e., direction of causality is not SES factors predicting race which would be required for SES factors to confound relationship of race and overdose fatality). Thus the partial model may be more relevant here. I would suggest stating both the partial and full model results for race in your results and interpreting some of this in the discussion because it suggests that some attention to structural factors disproportionately affecting African Americans could decrease opioid overdose mortality among African Americans. Presenting only the full model results of a protective relationship suggests that African American communities are less affected by the opioid epidemic, which may not necessarily be the full story here.

11. Line 248: Might consider making the statement “patients using opioid therapy for chronic pain should be carefully selected” a bit more specific and actionable by citing CDC opioid prescribing guidelines (Dowell et al CDC Guideline for Prescribing Opioids for Chronic Pain--United States, 2016). Could say something like: “physicians should weigh the benefits and risks of opioid therapy using national guidelines for opioid prescribing.” More recently these guidelines have been evaluated and you could also evaluate that literature in rewriting this sentence.

7. PLOS authors have the option to publish the peer review history of their article (what does this mean?). If published, this will include your full peer review and any attached files.

Reviewer #1: Yes: Ju Nyeong Park

Reviewer #2: No

---

## [Author Response · Author response to Decision Letter 1]

6 Dec 2019

Scientific Editor:

Comment 1. As you revise your manuscript according to the minor revisions suggested by the two reviewers, I also encourage you to give some additional thought to the relationship between race, SES and fatal overdose.

Response: This essential comment was address in the (new) second paragraph of the discussion on lines 252-254. Please see detailed response to Reviewer 2, Comment 10.

Comment 2. As reviewer #2 has suggested, consider presenting the partially adjusted results in light of the complex relationships between SES and race. 

Response: This excellent suggestion was adopted in the results on lines 194-196. Please see response to Reviewer 2, Comment 10 below.

Reviewer #1: 

Comment 1. Abstract: Refrain from using stigmatizing language. Replace “disabled people” with “people with disabilities”

Response: Throughout the manuscript we have attempted to adopt this suggestion.

Comment 2. Introduction: The authors have done a nice job of summarizing the literature on the opioid overdose epidemic. An explanation of the gaps and limitations in the literature would round out this section so that the reader can understand the novelty of the current study.

Response: In the introduction, from line 71 to 78, we have included this explanation:

“Although data on SES attributes including education, income, and employment are available at the county [11] and census tract-level [12], the gold standard for analysis is use of individual-level data to examine effects of personal attributes. There is a paucity of individual-level data on prospective relationships between individual-level SES measures and risk of fatal opioid overdose, including for critical factors such as health insurance coverage, employment and marital status, and incarceration. [13] 

National surveillance systems for opioid mortality typically do not capture detailed individual-level SES data. [1, 5, 6, 7, 8] Well-designed studies that include these data are often set in smaller geographic areas such as states, [10] and are not generalizable to the U.S. population.”

Comment 3. Results: Given the large sample size, I suggest reporting one decimal place for all percentages, both in the text and the tables.

Response: This recommendation was followed in both Table 1 and the results text.

Comment 4. In table 2, level of statistical significance would likely be of interest to readers. I would indicate using footnotes (e.g., * p <0.05, ** p<0.001 etc.)

Response: This recommendation was accepted in revision of Table 2.

Comment 5. Discussion: Another limitation is that medication and behavior therapy coverage was not measured. The authors may want to mention this fantastic consensus report that highlights national gaps in coverage https://www.nap.edu/catalog/25310/medications-for-opioid-use-disorder-save-lives

Response: The recommendation is used in new paragraph of the discussion, with the inclusion of reference 32, lines 259-275:

“People who were disabled had almost three times higher risk of death from opioid overdose than those without a disability, likely reflecting use of opioid analgesics to treat chronic pain. In 2016, CDC published guidelines to assist prescribers in weighing the benefits and risks of opioid therapy for chronic pain. [29] In 2019, the guidelines were evaluated by a consensus panel [30] and the CDC published a perspective [31] on measures to prevent misapplications of the guidelines that can cause harm. Another consensus report highlighted national gaps in evidence-based coverage that can save lives. [32] Examples include inflexible application of dosage and duration thresholds, abrupt tapering of opioid dosages, drug discontinuation, or dismissal of patients from care. Misapplication of the guidelines to other patient populations is another concern. This includes patients with pain at end-of-life, from cancer, acute surgical recovery, or sickle cell crises. Application of chronic pain dosage guidelines when prescribing opioid agonists to treat opioid use disorder can also cause harm. A need exists for empathetic chronic pain management such that non-opioid treatment is provided to the need for opioids, while taking into consideration the risks associated with each type of treatment. When patients agree to taper the dose of opioids, it is helpful for the pace to be individualized and gradual, to minimize withdrawal symptoms. [32] Further research on alternative chronic pain management strategies could point to interventions that lower opioid overdose mortality among patients at risk for opioid use disorder because of their medical comorbidities. [33]”

Comment 6. Minor comments, Abstract:

Replace “individual” with “individual-level” in the second sentence of the abstract.

Response: Lines 1-2 of the abstract were revised to incorporate this suggestion:

“Understanding relationships between individual-level demographic, socioeconomic status (SES) and U.S. opioid fatalities can inform national interventions in response to this crisis.”

Comment 7. In the results, “those with the highest level of educational attainment of high school” does this include college? This finding, as written, does not standalone.

Response: Table 1 and 2 now include the category “High School/GED only.” On line 177-179, this sentence was revised:

“People whose highest level of educational attainment was a High School diploma or GED only, with no college accounted for…”

On line 216 similar terminology is used:

“HRs among those with attainment of a High School diploma or GED only…”

Reviewer #2: 

Comment 1. Throughout: use person-first language where you are able. For example, line 68: “the homeless, and recently released prisoners” could be changed to persons who have experienced homelessness, housing instability, or incarceration. Also line 285 “substance users” should be changed to persons who use substances

Response: The manuscript was revised to use person-first language throughout. On line 308 “substance users” was revised to “people who use substances.”

Comment 2. Line 24: “Opioid fatalities are a U.S. problem.” – I think the prior version’s topic sentence was stronger and you might remove this sentence in favor of expanding the second sentence’s justification for the analysis.

Response: The sentence beginning on line 24 was revised as suggested:

“Understanding relationships between individual-level demographic, socioeconomic status (SES) and U.S. opioid fatalities can inform national interventions in response to this crisis.”

Comment 3. Line 29: Likewise “HRs for fatal opioid overdose by SES in final model:” in the results section is not needed. Clearer to go straight to the results you interpret.

Response: The abstract was revised as recommended on line 30-31:

“In multivariable analyses, compared to…”

Comment 4. Line 41: “West North Central state”, do you mean “West, North, or Central”? Potentially missing a word or were these areas combined?

Response: The “West North Central” States are one Census district that includes Missouri, Kansas, Iowa, Nebraska, the Dakota, Minnesota, and Wisconsin. Due to abstract word limits, this terminology was retained. 

Comment 5. Lines 45-46: Could strengthen the conclusions by suggesting how these findings from MDAC inform prevention, treatment, and rehabilitation (e.g. what communities or regions need more services?)

Response: Within constraints of abstract word limits the following revision was made on lines 45-46:

“ The findings may help to target prevention, treatment and rehabilitation efforts to vulnerable groups.”

Comment 6. Introduction lines 69-71: “Some risk factors such as unemployment and health insurance status are optimally studied using individual-level longitudinal data, with adjustment for other covariates. [11]” I do not get a lot of information from this sentence about what is known about the relationship of employment and health insurance coverage with opioid overdose. Could you instead highlight the main finding(s) of citation 11?

Response: Lines 71-75 were revised to indicate that individual-level data is the gold standard for analyses of SES: 

“Although data on SES attributes including education, income, and employment are available at the county [12] and census tract-level [13], the gold standard for analysis is use of individual-level data to examine effects of personal attributes. There is a paucity of individual-level data on prospective relationships between individual-level SES measures and risk of fatal opioid overdose, including for critical factors such as health insurance coverage, employment and marital status, and incarceration. [14] 

Comment 7. Introduction, lines 77-78: “after multivariable adjustment” – It may be clearer to say you hypothesized these would remain independently associated with fatal opioid overdose even after adjustment for other socioeconomic and demographic indicators (or something similar). ‘Multivariable adjustment’ on its own leaves the reader unclear on what you are doing.

Response: This comment was integrated with authors instruction for the introduction, to “conclude with a brief statement of the overall aim of the work and a comment about whether that aim was achieved.” Lines 82-83 were revised as follows:

“The MDAC database supported our aim to estimate hazard ratios for demographic, geospatial, and individual-level SES risk factors and fatal opioid overdose in the United States.”

 

Comment 8. Methods, lines 93-95: “Representativeness of the sample relative to the United States population was enhanced by applying weights to account for variable sampling.” – I don’t think you need this sentence and it makes the description of sampling weights less clear. The phrase ‘variable sampling’ is vague – unclear if variable is used as an adjective (as in varying) or noun (as in age, sex, etc. variables used for sampling).

Response: When lines 94-101 were revised, addressing sample weights, this sentence was deleted:

“The sample frame for the ACS is derived from the Master Address File. Sampling is designed to approximate age, sex, race, Hispanic ethnicity, and state of residence distributions observed in the Census Bureau’s annual United States population estimates. Whites were oversampled by about 2% compared to the 2010 Census, and therefore observations for this group were assigned weights of slightly less than one. Other racial/ethnic groups were undersampled by one percent or less, given weights slightly above one. Overall weights were centered on one observation per respondent i.e., the weighted sample approximates the number of 2008 ACS responses rather than the U.S. population. Table cell counts were rounded to four significant digits to prevent disclosure of identity.”

Comment 9. Results, line 156: “weighting to the U.S. population” could be clearer if you say “weighting to the age, sex, etc…distribution of the U.S. population.” Otherwise wording could imply that the sample size is weighted to the size of the US population and someone reading quickly or without much background in survey data analysis might misunderstand that only 3800 opioid overdoses occurred in the US if not clarified that the exact numbers do not represent the US population size.

Response: Lines 148-149 was revised as follows:

“After… weighting to age, sex, race, Hispanic ethnicity, and state of residence distributions of the U.S. population…”

Comment 10. The result in the final model that black participants were at lower risk of fatal overdose should be considered more thoroughly. The unadjusted and partially adjusted models suggest black race is a risk factor for fatal overdose and this association only becomes protective after adjusting for many SES related factors that disproportionately affect African American communities. A model considering race as a main exposure should not be adjusted for SES factors because race would predict some of those SES indicators but the SES factors would be mediators of relationships between race and opioid fatality (i.e., direction of causality is not SES factors predicting race which would be required for SES factors to confound relationship of race and overdose fatality). Thus the partial model may be more relevant here. I would suggest stating both the partial and full model results for race in your results and interpreting some of this in the discussion because it suggests that some attention to structural factors disproportionately affecting African Americans could decrease opioid overdose mortality among African Americans. Presenting only the full model results of a protective relationship suggests that African American communities are less affected by the opioid epidemic, which may not necessarily be the full story here.

Response: In the results, on lines 194-196, the less adjusted model results are described: 

“ In univariate and partially adjusted models Blacks had increased hazard rates (HR=1.37, CI: 1.16, 1.60) and (HR=1.44, CI: 1.22, 1.69), respectively, although those rates became lower than 1 after further adjusting for SES in the final model (HR=0.81, CI: 0.68, 0.96).”

Also, in the paragraph 2 of the discussion (lines 243-254) we address this important comment:.

“If SES partially mediated the effect of race in the final model, then interventions that impact SES, such as improving education, may have among their many advantages, the ability of help decrease opioid overdoses and associated racial disparities. [25-28]”

Comment 11. Line 248: Might consider making the statement “patients using opioid therapy for chronic pain should be carefully selected” a bit more specific and actionable by citing CDC opioid prescribing guidelines (Dowell et al CDC Guideline for Prescribing Opioids for Chronic Pain--United States, 2016). Could say something like: “physicians should weigh the benefits and risks of opioid therapy using national guidelines for opioid prescribing.” More recently these guidelines have been evaluated and you could also evaluate that literature in rewriting this sentence.

Response: Paragraph 3 of the discussion (lines 259-275) was expanded to include the CDC guidelines by Dowell et al (ref 29) and the Consensus panel report (ref 30) and CDC commentary (ref 31). A National Academies of Sciences, Engineering, and Medicine consensus report on opioid use disorder (ref 32) was also included at the recommendation of another reviewer of this manuscript:

“People who were disabled had almost three times higher risk of death from opioid overdose than those without a disability, likely reflecting use of opioid analgesics to treat chronic pain. In 2016, CDC published guidelines to assist prescribers in weighing the benefits and risks of opioid therapy for chronic pain. [29] In 2019, the guidelines were evaluated by a consensus panel [30] and the CDC published a perspective [31] on measures to prevent misapplications of the guidelines that can cause harm. Examples include inflexible application of dosage and duration thresholds, abrupt tapering of opioid dosages, drug discontinuation, or dismissal of patients from care. Misapplication of the guidelines to other patient populations is another concern. This includes patients with pain at end-of-life, from cancer, acute surgical recovery, sickle cell crises. Application of chronic pain dosage guidelines when prescribing opioid agonists to treat opioid use disorder can also cause harm. A consensus report highlights national gaps in evidence-based care for opioid use disorder that can save lives. [32] A need exists for empathetic chronic pain management such that non-opioid treatment is provided to the need for opioids, while taking into consideration the risks associated with each type of treatment. When patients agree to taper the dose of opioids, it is helpful for the pace to be individualized and gradual, to minimize withdrawal symptoms. [32] Further research on alternative chronic pain management strategies could point to interventions that lower opioid overdose mortality among patients at risk for opioid use disorder because of their medical comorbidities. [33]”

---

## [Editor Report · Decision Letter 2]

6 Jan 2020

Socioeconomic risk factors for fatal opioid overdoses in the United States: Findings from the Mortality Disparities in American Communities Study (MDAC)

PONE-D-19-16947R2

Dear Dr. Altekruse,

We are pleased to inform you that your manuscript has been judged scientifically suitable for publication and will be formally accepted for publication once it complies with all outstanding technical requirements.

With kind regards,

Becky L. Genberg

Academic Editor

PLOS ONE
---

## [Editor Report · Acceptance letter]

10 Jan 2020

PONE-D-19-16947R2 

Socioeconomic risk factors for fatal opioid overdoses in the United States: Findings from the Mortality Disparities in American Communities Study (MDAC) 

Dear Dr. Altekruse:

I am pleased to inform you that your manuscript has been deemed suitable for publication in PLOS ONE. Congratulations! Your manuscript is now with our production department. 

With kind regards,

on behalf of

Dr. Becky L. Genberg 

Academic Editor

PLOS ONE